# Silicon Wafer Etching Rate Characteristics with Burst Width Using 150 kHz Band High-Power Burst Inductively Coupled Plasma

**DOI:** 10.3390/mi12060599

**Published:** 2021-05-22

**Authors:** Hisaki Kikuchi, Katsuyuki Takahashi, Seiji Mukaigawa, Koichi Takaki, Ken Yukimura

**Affiliations:** 1Department of Systems Innovation Engineering, Faculty of Science and Engineering, Iwate University, Morioka, Iwate 020-8551, Japan; g0320041@iwate-u.ac.jp (H.K.); ktaka@iwate-u.ac.jp (K.T.); mukaigaw@iwate-u.ac.jp (S.M.); keny@gaia.eonet.ne.jp (K.Y.); 2Agri-Innovation Center, Iwate University, Morioka, Iwate 020-8550, Japan

**Keywords:** ICP, etching, HiPIMS, high power, burst pulse, double probe measurement, CF_4_, Ar

## Abstract

The high-speed etching of a silicon wafer was experimentally investigated, focusing on the duty factor of 150 kHz band high-power burst inductively coupled plasma. The pulse burst width was varied in the range of 400–1000 µs and the repetition rate was set to 10 Hz. A mixture of argon (Ar) and carbon tetrafluoride (CF_4_) gas was used as the etching gas and injected into the vacuum chamber. The impedance was changed with time, and the coil voltage and current were changed to follow it. During the discharge, about 3 kW of power was applied. The electron temperature and plasma density were measured by the double probe method. The plasma density in the etching region was 10^18^–10^19^ m^−3^. The target current increased with t burst width. The etching rate of Ar discharge at burst width of 1000 µs was 0.005 µm/min. Adding CF_4_ into Ar, the etching rate became 0.05 µm/min, which was about 10 times higher. The etching rate increased with burst width.

## 1. Introduction

Plasma processing, a material processing technique using plasma, is an essential part of today’s manufacturing industries. Plasma-based surface processes are used to manufacture micro and nano order high-density integrated circuits for the electronics industry, specifically, for plasma etching to remove unnecessary parts of the circuit substrate and for sputtering to form thin metal films in the semiconductor device manufacturing process [1,2,3]. Plasma etching is used in processes such as patterning, planarization, removal of damaged layers, and resist removal (ashing). Nowadays, ultrafine processing techniques are required to meet the demand for large-scale integrated circuits, and a high-density plasma source is necessary to achieve this [4]. Inductively coupled plasma (ICP) has been widely used as a plasma source because of its advantages, such as simple equipment configuration and high density plasma at low pressure [5,6,7,8,9]. ICP can be generated with electrodeless electrodes, and the positive ions can be independently controlled by the bias voltage applied to the substrate [5,6,7,8,10].

A new ICP system using burst waves has been developed [11,12,13,14,15,16,17]. In this system, the frequency is set to 100–200 kHz, which is not necessary for the impedance matching circuit required in conventional ICP systems, and the resonance between the ICP inductance and the parallel capacitance produces high-density plasma of the order of 10^19^ m^−3^ [12,13,14,15,16,17]. Therefore, this ICP system has the advantage of a simple equipment configuration. In fact, the plasma source is compact, with a cylindrical shape and about 50 mm in diameter, which costs less than conventional systems. In addition, the burst width and duty ratio can be freely set to optimize the process conditions. This plasma source can also be installed in a minimal fab system that processes half-inch wafers to produce semiconductor devices [18,19].

In this study, reactive ion etching was performed using a tetrafluoro-carbon (CF_4_) and argon (Ar) mixed gas atmosphere, and the etching rate was compared with that of physical etching using argon gas atmosphere. It has been reported that the etching rate of the burst wave ICP is affected by the substrate bias voltage, CF_4_ contents and input power [17]. In this study, the influence of the burst width on the electric and etching characteristics was investigated. The etching was performed on samples placed in the bulk plasma. The electrical characteristics were evaluated by time-resolved measurements. The burst width was varied in the range of 400–1000 µs. The target current and etching rate were evaluated at each burst width.

## 2. Experimental Procedure

Figure 1 shows a schematic of the experimental apparatus. The vacuum chamber consisted of a cylindrical glass tube (51 mm inner diameter, 55 mm external dimeter). The plasma source consisted of a solenoid coil (50 turns, length 70 mm, 83.7 μH) wound on a glass tube. A capacitor (12 nF) was connected to the coil in parallel and was used to build a resonance circuit with the coil. The capacitance and inductance were resonant with a frequency of about 157 kHz. The 400–1000 μs wide high voltage burst pulses with a frequency of 157 kHz were generated using a power supply (PS-1, PEKURIS KJ14-4873, Kyoto, Japan) and applied to the solenoid coil to generate ICP. The number of sine waves per burst pulse was 157 for the burst width of 1ms, and the repetition rate of the burst pulse was 10 Hz. Argon and tetrafluoro-carbon gases were supplied through mass flow controllers (SEC-400MK3 and SEC-E40MK3, HORIBA, Kyoto, Japan) into the chamber. Total gas flow rate was controlled in range of 19.3–41.7 sccm. The solenoid coil voltage and current were measured by a high voltage probe (P6015A, Tektronix, Beaverton, OR, USA) and a current monitor (model 110A, Pearson, London, UK), respectively.

Figure 2 shows the structure of the target electrode used in the etching process. The silicon wafer to be etched ((100), 12.6 mm diameter, 250 μm thickness) was set on the target electrode. A stainless steel mask (12.6 mm diameter, 150 μm thickness) with 1 mm square holes was placed on the wafer. The target electrode was made of titanium (26 mm diameter, 35 mm length) and had a cylindrical shape. The target electrode was placed inside the chamber. The distance between the target electrode and the end of the coil was 70 mm as shown in Figure 1. A negative-polarity rectangular pulse voltage with a pulse width of 1000 μs and a voltage of 800 V was applied to the target electrode by a power supply (PS-2, PEKURUS KJ06-3265, Kyoto, Japan) via a current limiting resistor of 5 Ω consisting of two 10 Ω resistors connected in parallel. The repetition rate was set to 10 Hz and the application timing was synchronized with the burst signal. During the first 4 min of etching, the presputter process was done by gradually increasing the bias voltage from about −100 V. The specified bias voltage was applied to the target for 20 min after the presputter process. After etching process, the etching depth of the wafer was measured by a surface roughness tester (Form Talysurf Super S5K, Taylor Hobson, Leicester, UK), and the etching rate was calculated by dividing the etching depth by the processing time [17]. The voltage applied to the target and current flowing through the electrodes were measured by a high voltage probe (P-5100, Tektronix) and current monitor (model 110A, Pearson), respectively.

The electron temperature and the ion density were obtained by floating double probe measurements [20,21,22,23]. The probe tip was a cylindrical tungsten electrode (ϕ 0.4) with an exposed length of 3 mm and a tip-to-tip distance of 8 mm. The measurement position of the probe was 70 mm from the coil edge. Electron energy distribution function was assumed as Maxwellian 2 [14,24].

## 3. Results and Discussion

Figure 3 shows the typical waveforms of the coil voltage and coil current, and the time evolution of the effective electrical power, the power factor, the impedance, the resistance and the reactance for pure Ar discharge and Ar/CF_4_ discharge. The total flow rates in Ar and Ar/CF_4_ discharge were Ar = 41.7 sccm and Ar/CF_4_ = 35.4/6 sccm, respectively, and the CF_4_ content rate was 15%. The pressure was 5 Pa. The burst width was 1000 μs and the repetition rate was 10 Hz. The duty ratio was 1%. The waveform was divided into 3 stages (I)~(III) as shown in Figure 3. The ICP in Ar/CF_4_ discharge was ignited at 40 μs (stage (II)). The peak values of coil voltage and current before the ICP ignition (stage (I)) were approximately 3.4 kV and 39 A, respectively. The impedance calculated from the amplitude of the coil voltage and current was about 87 Ω, which was almost equal to the impedance of the induction coil of 83 Ω. After the plasma ignition (stage (III)), the coil voltage and current decreased to 1.7 kV and 21 A, respectively. When the plasma was generated, a transformer, where the plasma was regarded as the secondary winding was coupled between the coil and the plasma, and the impedance changed. The impedance at this time was approximately 81 Ω [25,26]. After the time of 400 μs (stage (III)), the coil voltage and current increased again to approximately 2.2 kV and 25 A, respectively. This trend of time evolution was also observed in the Ar discharge. The amplitudes of the coil voltage and current were slightly larger when CF_4_ was included. When CF_4_ was added, the electron density decreased by electron attachment, and the plasma resistance, which was determined by collusions between electrons and other particles, was reduced. The power factor after the plasma generated was approximately 20%, and the effective power of the coil was approximately 4.5 kW. Therefore, the average power was 45 W at a repetition rate of 10 Hz.

The effective electrical power and the power factor for Ar and Ar/CF_4_ discharges were obtained by waveforms of coil voltage and current. The power factor was obtained from the phase difference at zero-cross timing between the voltage and current. In the Ar discharge, approximately 2 kW of power was consumed between 200–400 μs (32–63 cycle) in stage (II). After that, it increased to approximately 3.5 kW, and then became constant after 600 μs (90 cycle) in stage (III). The power factor increased rapidly from 0.0010 to 0.20 between 40–100 μs (6–6 cycles) in stage (I). After that, it gradually decreased to 0.21 in stage (II), and then decreases to 0.13, and then has a constant value in stage (III). In the Ar/CF_4_ discharge, approximately 3 kW of power was consumed during the discharge. The power factor increased rapidly from 0.0029 to 0.16 between 40–130 μs (6–20 cycles) in stage (I). After that, it gradually increased to 0.19 at 200 μs (32 cycle), and then decreased to 0.10 at 580 μs (94 cycle). After that, it was almost constant in stage (III).

In the Ar discharge, the impedance decreased rapidly from 88.0 Ω to 80.9 Ω between 40–110 μs (6–16 cycle) in stage (I), and it increased to 84.1 Ω in stage (II). After that, it decreased again to 82.7 Ω at 530 μs (80 cycle), and it increased gradually to 87.4 Ω in stage (III). This trend of time evolution was also observed in the resistance and the reactance. The resistance had a symmetrical trend, and the reactance had the same trend as the impedance. In the Ar/CF_4_ discharge, the impedance decreased rapidly from 87.8 Ω to 82.3 Ω between 40–130 μs (6–19 cycle) in stage (I), and it decreased to 80.8 Ω in stage (II). After that, the impedance gradually increased to 87.1 Ω in stage (III). The impedance change had the same timing as the coil voltage and current. The resistance increased rapidly from 0.25 Ω to 13.1 Ω between 40–130 μs in stage (I). The resistance increased in the number of electron collisions with other particles in the plasma [26,27,28,29,30]. Thus, the electron density increased rapidly during this period. After that, it increased slightly to 15.2 Ω, and it was almost constant until 400 μs in stage (II). After 400 μs, the resistance component decreased at 600 μs in stage (III). The loss process of electrons was larger than the generation of electrons in this period. After 600µs, the resistance was almost constant. The reactance change timing was the same as that of the resistance. Because the reactance decreased between 40–130 μs, and it increased until 600 μs, and it was almost constant.

Figure 4 shows the effective values of the coil voltage and current for Ar discharge, Ar/CF_4_ discharge, effective power and power factor as a function of input power. The input power was the time average of the power to the coil during the discharge. The coil voltage and current increased with the input power. The values of coil voltage and current in Ar and Ar/CF_4_ discharges were almost equal. The effective power was almost equal for Ar and Ar/CF_4_ discharges, ranging from about 3.0–11 kW. It was found that it increased linearly with the input power. The power factor increased with input power and it was 0.19–0.28. The power factor in Ar/CF_4_ discharge was smaller than the Ar discharge.

Figure 5 shows the impedance (Z), resistance (R), and reactance (X) during the discharge as functions of the input power. The impedance decreased with increasing input power. The impedance of Ar/CF_4_ discharge was larger than that of the Ar discharge. The resistance increased, and the reactance decreased with increasing input power. The resistance during discharge was 15–21 Ω for Ar discharge and 15–20 Ω for Ar/CF_4_ discharge, which was larger than the parasitic resistance of the coil, 0.16 Ω. The reactance of the Ar/CF_4_ discharge was higher than that of the Ar discharge.

Figure 6 shows the electron temperature (*T*_e_) and plasma density (*n*_p_) of Ar and Ar/CF_4_ discharges as functions of the input power. The target electrode was not placed in the chamber. The probe measurement position was 70 mm from the coil edge. The ion density *n*_i_ is obtained as following equation [31],
(1)$$n_{i}=J/0.61eu_{B}$$
where *J* is the target current density obtained from the measured target current *I*_t_, *u*_B_ is Bohm velocity and *e* is an elementary charge. The electron temperature is 2–3 eV and is not affected by gas species. In conventional ICP, the electron temperature is around 3.5 eV [9,32,33,34], and this ICP is almost equal to the conventional value. The plasma density increases with the input power, and is on the orders of 10^19^ m^−3^ in Ar discharge, and 10^18^ m^−3^ in Ar/CF_4_ discharge. In conventional ICP, the plasma density is on the orders of 10^17^–10^18^ m^−3^ in Ar discharge, and 10^16^ m^−3^ in Ar/CF_4_ discharge [1,5,35]. In this ICP, it was 10 to 100 times higher than the conventional ICP. Therefore, it is expected to be applied to high-speed processes. The plasma density in the Ar/CF_4_ discharge is lower than that in Ar discharge because of electron loss due to electron attachment to various kinds of molecules [35,36,37,38]. When an electronegative gas such as CF_4_ is fed into the chamber, the potential structure in the sheath has two-stages. In this case, the negative ions are confined to the center of the plasma, and the positive ion current flowing through the electrode is proportional to the electron density, not to the positive ions. Therefore, the value of the obtained saturated ion current is different from that for positive ions only. Assuming that only a single negative ion species is presented in the plasma, and it is lost by the recombination with positive ions, the balance equation can be expressed as follows,
(2)$$k_{a}n_{e}N_{0}=k_{r}N_{p}N_{n}\approx k_{r}N_{n}^{2}$$
where *k*_a_ is the attachment rate coefficient, *k*_r_ is the recombination coefficient, *n*_e_ is electron density, *N*_p_ is positive ion density, *N*_n_ is negative ion density, and *N*_0_ is the reaction gas density (in this case, CF_4_ density). From the quasi-neutral condition with *n*_p_ ≈ *n*_n_, the ratio of the negative ion density t the electron density, *β*, in Equation (2) becomes,
(3)$$\beta =\sqrt{k_{a}N_{0}/k_{r}n_{e}}\;$$

Assuming the negative ion is only F^−^ and the electron temperature is 3 eV, *k*_a_, *k*_r_ and *β* are estimated as 8.9 × 10^−17^ m^3^s^−1^, 4.0 × 10^−13^ m^3^s^−1^, and 0.065, respectively [36,37,39]. Therefore, the negative ion density is 7.15 × 10^17^ m^−3^. Because the negative ion density is much smaller than the electron density, the negative ion does not affect the saturated ion current flowing into the probe.

Figure 7 shows the target current for Ar discharge and Ar/CF_4_ discharge for each burst width at a pressure of 5 Pa. The target position was 70 mm from the coil edge. The bias voltage was −800 V, and the pulse width was synchronized with the burst width. In the Ar discharge with a burst width of 1000 μs, the target current rose from approximately 130 μs. The target current reached a maximum value of about 2.5 A at 600 μs. The voltage drop across the current control resistor (5 Ω) was a few tens of volts and was sufficiently low to the bias voltage. In the Ar/CF_4_ discharge, the target current gradually rose from approximately about 150 μs and reached a maximum value of approximately 1.1 A at 650 μs. The maximum current value was approximately half of that of the Ar discharge. The target current decreased significantly and the current rise time was delayed when the CF_4_ was fed in. Because CF_4_ was fed into the chamber, the ion density decreased and the target current decreased. The burst width increased, the maximum value of the target current became slightly larger, and the rise time became shorter. The density of high-energy electrons increased with burst width.

Figure 8 shows the target current for Ar discharge and Ar/CF_4_ discharge at each input power. Ar = 19.3 sccm, Ar/CF_4_ = 14.4/5 sccm, for the pressure of 3 Pa. The burst width was 400 μs, the repetition rate was 25 Hz, and duty rate was 1%. The target position was 70 mm from the coil edge. The bias voltage was −800 V, and the pulse width was synchronized with the burst width. The burst width was set to 400 µs in order to reduce the arcing at the target electrode and make the process stable. The pressure was changed from 5 to 3 Pa by reducing the gas flow rate from 41.7 to 19.3 sccm also, in order to inhibit the arcing. As noted in Figure 9, the density of the Ar discharge was of the order of 10^19^ m^−3^ and the Ar/CF_4_ discharge was of the order of 10^18^ m^−3^. The input power increased, the maximum value of the target current became larger, and the time to ignite plasma and the rise time of the target current became shorter. The high-energy electrons increased with input power, which accelerated the ionization. When CF_4_ was added, the EEDF showed a decrease in the fraction of low-energy electrons and an increase in the fraction of high-energy electrons [36,38]. This suggests that the generation process of Ar ions is dominated by the multistep ionization of Ar atoms in the Ar discharge, and direct ionization from the ground state increases in Ar/CF_4_ discharge. Therefore, the discharge is ignited earlier by the addition of CF_4_. In addition, a large change in the target current was observed at 3.0 kW and 3.9 kW in the Ar discharge. As shown in Figure 6, the plasma density in Ar/CF_4_ discharge was smaller than the Ar discharge at low input power, because the attachment rate of electrons in CF_4_ is larger than the ionization rate in Ar at low input power, i.e., a low reduced electric field (E/N) [10]. Therefore, the plasma density decreases with adding CF_4_ gas into Ar gas at low input power. However, the ionization rate in Ar is much larger than the attachment rate of electrons in CF_4_ at high input power, i.e., a high reduced electric field (E/N) [10]. Therefore, the plasma density in the Ar/CF_4_ discharge had almost same value as that in the Ar discharge at high input power, as shown in Figure 6b. The plasma impedance is affected by the plasma density. The change of plasma impedance is one of the reasons for the target current fluctuation shown in Figure 8. However, more study is needed to clarify the detail of the mechanism.

Figure 9 shows the etching rate for Ar discharge and Ar/CF_4_ discharge as a function of the input power. The etching rate by Ar discharge was much smaller than Ar/CF_4_ discharge. Because in the etching in Ar discharge, the Ti target electrode was etched by argon ions, and fractions were deposited in the trench [17]. The etching rate in Ar/CF_4_ discharge increased with input power, and the maximum etching rate was 0.13 µm/min. In conventional ICP, the etching rate is 0.02–0.03 μm/min [40]. In this ICP, the etching rate was faster than the conventional ICP. This is supported by the high density compared to the conventional ICP.

Figure 10 shows the etching rate for Ar discharge and Ar/CF_4_ discharge as a function of the burst width with a repetition rate of 10 Hz. The repetition rate was fixed, the input power per cycle was increased with burst width. Ar = 41.7 sccm, Ar/CF_4_ = 35.4/6 sccm, and the pressure was 5 Pa. In Ar discharge, the etching rate increased slightly with the increasing burst width and was about 0.005 μm/min. In Ar/CF_4_ discharge, the etching rate increased slightly with the increasing burst width and was about 0.02–0.05 μm/min. The etching rate in the case of Ar/CF_4_ discharge was 4–10 times higher than that of Ar discharge.

Figure 11 shows the etching rate for Ar discharge and Ar/CF_4_ discharge as a function of the burst width with a duty cycle of 1.2%. The duty cycle was fixed, the input power per cycle did not change. When the duty ratio was fixed, there was no significant change with the burst width, unlike when the repetition rate was fixed. Therefore, the etching rate is affected by the input power. The etching rate was about 0.005 µm/min for Ar discharge and 0.04–0.05 µm/min for Ar/CF_4_ discharge.

## 4. Conclusions

The electrical characteristics and etching rates of the high-power burst Ar and Ar/CF_4_ ICPs were investigated. The effective power, power factor and impedance were obtained by time-resolved measurements. The etching was operated in Ar and Ar/CF_4_ discharges, and the effects of the power and burst width were investigated. It was found that during Ar/CF_4_ discharge, about 3 kW was applied. It was confirmed that the coil current and voltage changed with the impedance. The power factor in Ar/CF_4_ discharge was slightly smaller than Ar discharge, and the impedance was larger than the Ar discharge. Double probe measurements showed that the plasma density in the etching area was in the order of 10^18^–10^19^ m^−3^, that the etching was performed under high density. The etching rate of the silicon wafer by Ar/CF_4_ discharge (reactive ion etching) was larger than the Ar discharge (physical etching) and increased 0.05 µm/min with the burst width increasing at a constant repetition rate. At this time, since the duty ratio is 1.0%, a DC equivalent value of 5 µm/min could be obtained.

## Figures and Tables

**Figure 1 micromachines-12-00599-f001:**
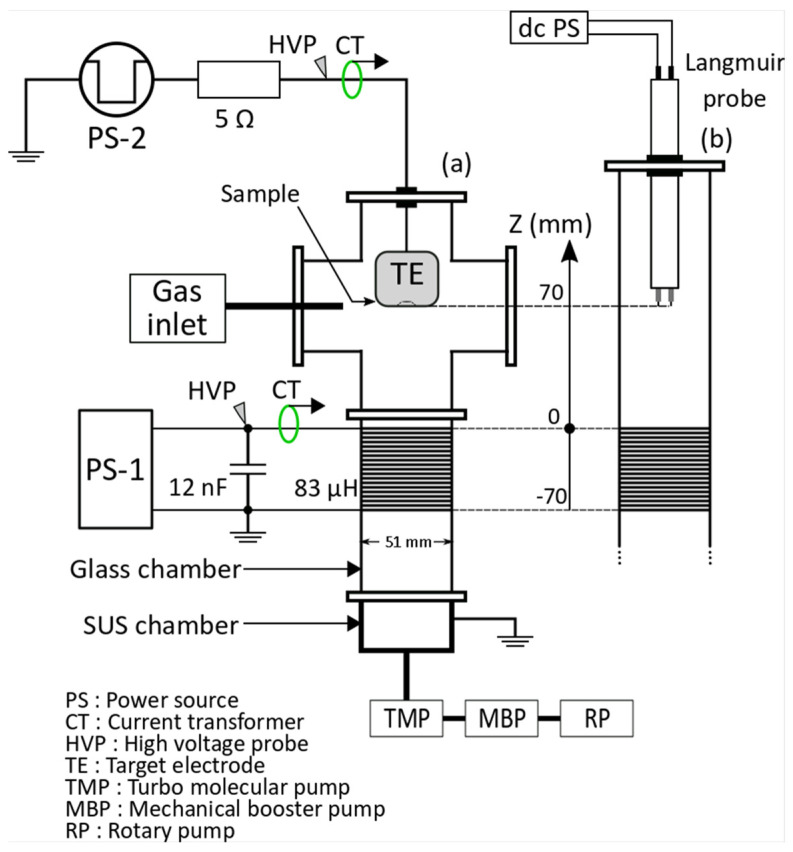
Schematic of the experimental apparatus.

**Figure 2 micromachines-12-00599-f002:**
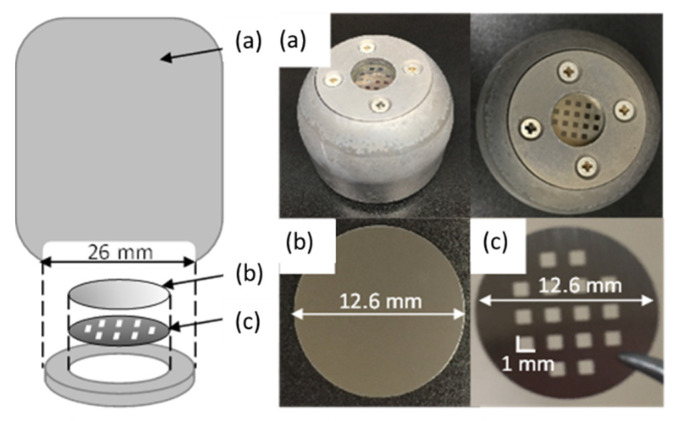
Structure of target electrode. (**a**) Target electrode, (**b**) silicon wafer, and (**c**) stainless steel mask.

**Figure 3 micromachines-12-00599-f003:**
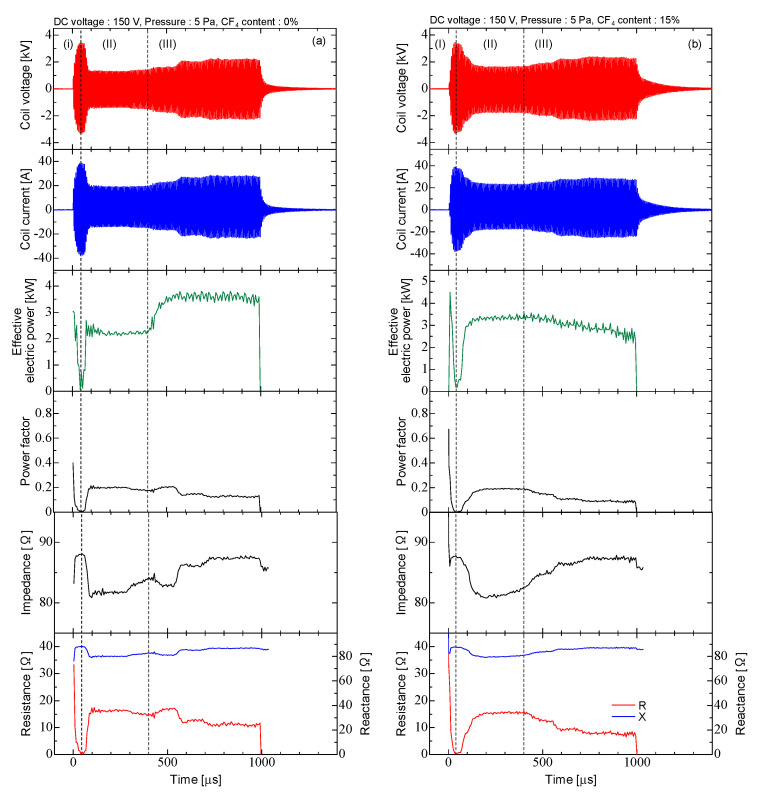
Typical waveforms of the coil voltage, coil current, the time evolution of the effective electrical power, the power factor, the impedance, the resistance and the reactance. (**a**) Ar discharge and (**b**) Ar/CF_4_ discharge (**I**) before discharge, (**II**) constant discharge, (**II**) discharge.

**Figure 4 micromachines-12-00599-f004:**
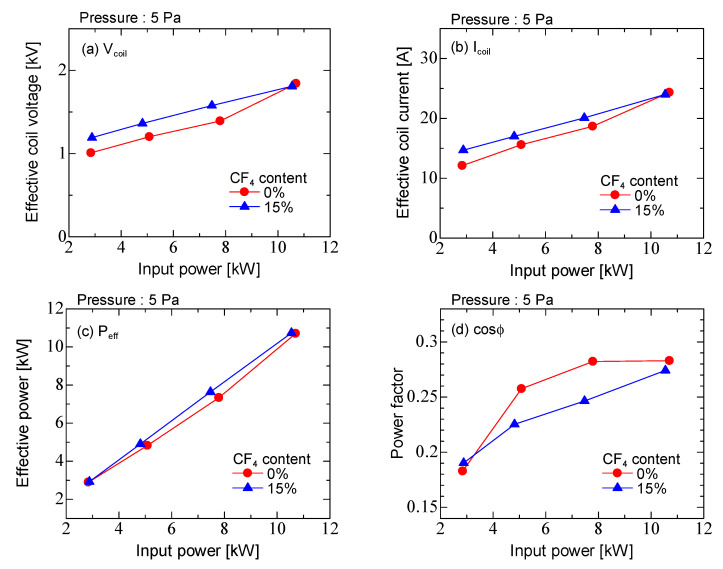
Electrical characteristics of the ICP as a function of input power for Ar discharge and Ar/CF_4_ discharge. (**a**) Effective coil voltage, (**b**) effective coil current, (**c**) effective power and (**d**) power factor.

**Figure 5 micromachines-12-00599-f005:**
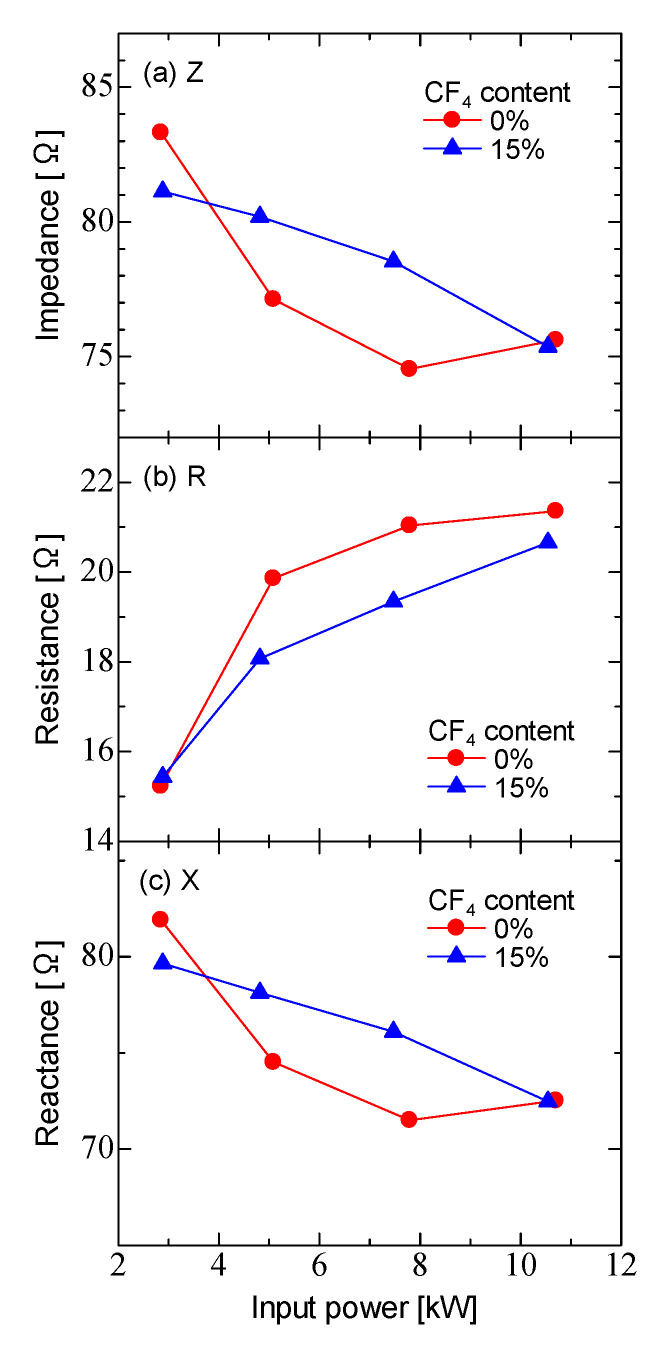
Electrical characteristics of the ICP as a function of input power for Ar discharge and Ar/CF_4_ discharge. (**a**) impedance, (**b**) resistance and (**c**) reactance.

**Figure 6 micromachines-12-00599-f006:**
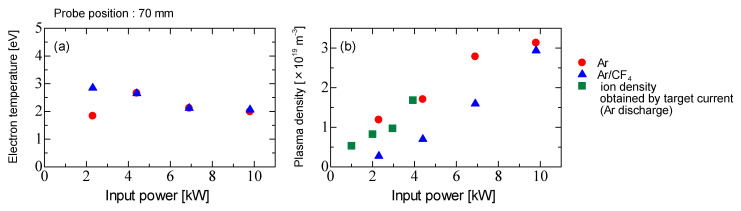
Plasma parameters as a function of input power. (**a**) Electron temperature and (**b**) plasma density for Ar and Ar/CF_4_ discharges.

**Figure 7 micromachines-12-00599-f007:**
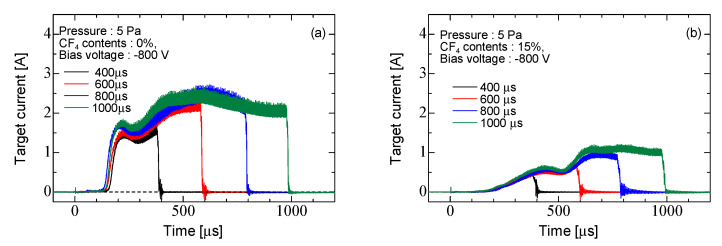
Waveform of target current at each burst width (**a**) Ar discharge and (**b**) Ar/CF_4_ discharge.

**Figure 8 micromachines-12-00599-f008:**
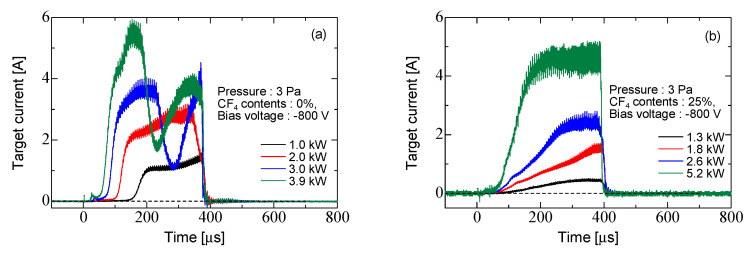
Waveform of target current at each input power. (**a**) Ar discharge and (**b**) Ar/CF_4_ discharge.

**Figure 9 micromachines-12-00599-f009:**
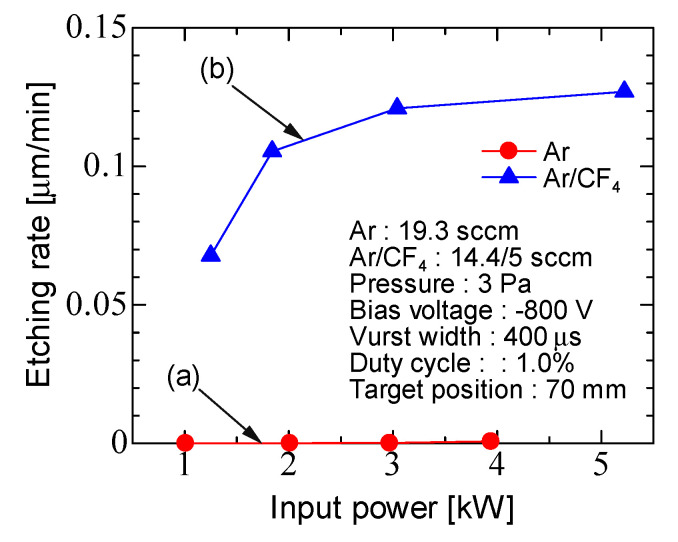
Input power characteristics of the etching rate for (**a**) Ar discharge and (**b**) Ar/CF_4_ discharge.

**Figure 10 micromachines-12-00599-f010:**
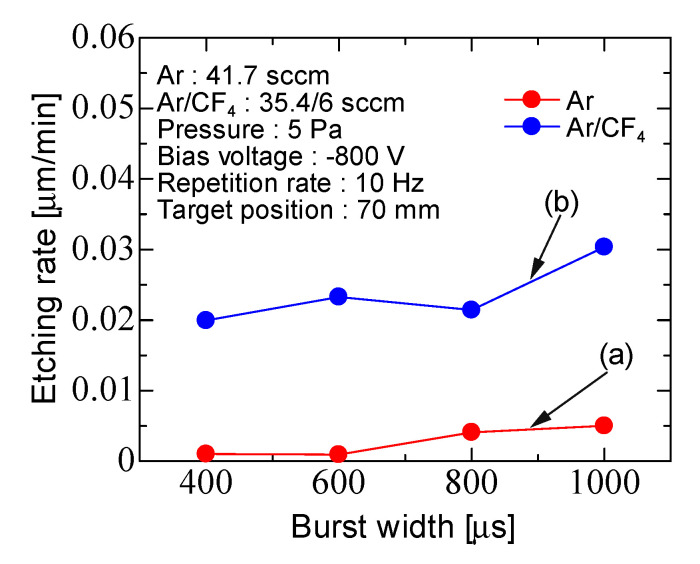
Burst width characteristics of the etching rate for (**a**) Ar discharge and (**b**) Ar/CF_4_ discharge with fixed repetition rate (10 Hz).

**Figure 11 micromachines-12-00599-f011:**
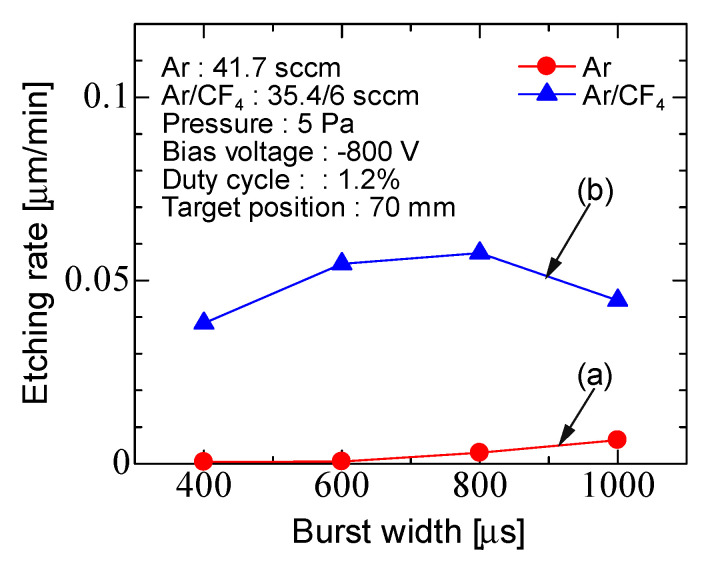
Burst width characteristics of the etching rate for (**a**) Ar discharge and (**b**) Ar/CF_4_ discharge with fixed duty cycle (1.2%).

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
