# Peer review of "Silicon Wafer Etching Rate Characteristics with Burst Width Using 150 kHz Band High-Power Burst Inductively Coupled Plasma"

_micromachines, 2021, doi:10.3390/mi12060599_

Round 1

Reviewer 1 Report

The authors reported the Si wafer etching characteristics using a 150 kHz Band High-Power Burst ICP system in Ar and Ar/CF4 gas. However, comparing Si etching with Ar and Ar/CF4 gas is a general result, and the advantages of the plasma system claimed by the authors are not shown.

The authors need to more clearly demonstrate the properties of the plasma source in order to be published on Micromachines.

(1) Explain in the manuscript what the 150 kHz Band High-Power Burst ICP system is superior to the conventional 13.56 MHz ICP system.

(2) The characteristics of the plasma system claimed by the authors should be compared with the conventional ICP system. At least compare the ion density, electron temperature, and etch rate.

Reviewer 2 Report

In this manuscript, authors carried on a research of high power burst Ar and Ar/CF4 ICP, where the electrical characteristics, (e.g. power and impedance) and etching rates were measured and compared. In general, this manuscript is quite well written, with experiments described clearly and results presented in a proper way. The aim of improving the chemical etching rate and stability is of great significance in the semiconductor industries, and is interested by a wide range of audience. I would recommend this high quality piece of research published with minor improvements suggested or questions to be addressed as below:

  1. The total gas flow rate is controlled range of 30-39 sccm (line 56). Why the flow rate in results section is either 19.3 or 41 sccm (out of the range)?
  2. Only etching rate was discussed in this manuscript, while the authors prepared to measure the surface roughness after etching process (line 69). how about the surface quality? I understand this might not be the focus of this article, but feel it would mislead readers if it was mentioned in experimental preparation while no results mentioned.
  3. It is unclear in the Section 3 why authors changed the experimental parameters from 41 sccm and 5 pa to 19. sccm and 3 pa. I would suggest on paragraph to be added to elaborate this change and necessary explanations of the objective.
  4. In Section 3, I feel authors did quite well in presenting the results, but there is lack of discussions of scientific findings or explanations behind the results. e.g. in Figure 8, clearly there are huge fluctuations of target current at 3.0 kW and 3.9 kW, why?

Overall, I think the manuscript is of good quality in experimental methods, data acquisition, graphic and written communications. 

Round 2

Reviewer 1 Report

Manuscript is well revised

I recommend this manuscript to be accepted by Micromachines.